# Synthesis of Hydantoin Androgen Receptor Antagonists and Study on Their Antagonistic Activity

**DOI:** 10.3390/molecules27185867

**Published:** 2022-09-10

**Authors:** Longjun Ma, Yan Zhou, Dehua Yang, Ming-Wei Wang, Wei Lu, Jiyu Jin

**Affiliations:** 1Department of Applied Chemistry, School of Chemistry and Chemical Engineering, Lanzhou Jiaotong University, 88 Anning West Road, Lanzhou 730070, China; 2Shanghai Engineering Research Center of Molecular Therapeutics and New Drug Development, School of Chemistry and Molecular Engineering, East China Normal University, 3663 North Zhongshan Road, Shanghai 200062, China; 3The National Center for Drug Screening, CAS Key Laboratory of Receptor Research, Shanghai Institute of Materia Medica, Chinese Academy of Sciences (CAS), 189 Guo Shou Jing Road, Shanghai 200031, China

**Keywords:** androgen receptor, antagonist, enzalutamide, hydantoin analogs, antiproliferative

## Abstract

Hydroxymethylthiohydantoin, hydroxymethylthiohydantoin, and hydantoin, containing a pyridine group, were synthesized to study their androgen receptor antagonistic activities. Among them, compounds **6a**/**6c**/**7g**/**19a**/**19b** exhibited excellent androgen receptor antagonistic activity, which was consistent with or even superior to enzalutamide. In addition, compounds **19a** and **19b** exhibited better antiproliferative activity than enzalutamide in prostate cancer cells. The results show that compound **19a** has great potential as a new AR antagonist.

## 1. Introduction

Prostate cancer, fueled by the androgen axis, is a major public health problem and has a very high leading cause of death among men worldwide [1]. Although androgen deprivation therapy (ADT) has been proved to be effective initially, the tumor will eventually progress and develop into the lethal castration-resistant prostate cancer (CRPC) [2]. Most often, death occurs 2 to 4 years after the onset of the castration-resistant state. However, the over-expression of AR was found in most CRPCs, which is essential for CRPC to adapt to the low levels of androgen. As AR receptor activation plays a crucial role in the progression of CRPC, it has been recognized as an attractive target for the treatment of CRPC [3].

AR antagonists are currently the mainstay of treatments for prostate cancer [4]. Bicalutamide, enzalutamide and apalut salts are marketed as nonsteroidal antiandrogens [5,6,7]. As a first-generation nonsteroidal AR antagonist, bicalutamide diminishes androgenic effects by competitively inhibiting androgen–AR binding. However, due to the occurrence of LBD point mutations and the expression of active AR splice variants, these antiandrogens may become partial AR agonists after a period of treatment (~2 years) [5]. (Figure 1).

Enzalutamide and apalutamide (ARN-509) are second-generation nonsteroidal antiandrogens with high-affinity binding for the AR LBD. In particular, enzalutamide received FDA approval in 2012 for the treatment of patients with metastatic castration-resistant PC who have previously received docetaxel [8,9]. However, a F876L missense mutation in the AR LBD has been shown to confer resistance to enzalutamide and apalutamide (ARN-509) by switching their activity on ARs from antagonist to agonist [10,11]. (Figure 1).

Enzalutamide and apalutamide (ARN-509) are very similar in structure and function to each other. The two agents are potent AR antagonists, which both have a high affinity for the AR. Both of them bind to AR and inhibit androgen-mediated gene transcription in AR-overexpressing prostate cancer cells, but also impair the nuclear localization of AR and DNA binding. However, in murine xenograft models of mCRPC, apalutamide demonstrated greater antitumor activity than enzalutamide. Besides that, apalutamide penetrates less effectively in the BBB (blood–brain barrier) than enzalutamide, suggesting that the chance of developing seizures may be less than with enzalutamide [12,13].

In this work, enzalutamide was used as the lead compound for structural modification. The structural transformation is as follows: (1) the aromatic ring on one side of hydroxymethyl thiohydantoin was transformed to contain both *p*-CN and *m*-CF_5_, and an active derivatized group of enzalutamide was connected to the aromatic ring on the other side, which has been proven to have good properties [14,15,16]; (2) the sulfur atom in the hydroxymethylthiohydantoin was replaced by an oxygen atom, which was shown to have unexpected effects in our previous work [17]; (3) the aromatic ring on the right side of the hydroxymethylthiohydantoin ring was replaced by a pyridine ring, which also contains both *p*-CN and *m*-SF_5_. A total of 30 compounds with CF_3_ groups were designed and synthesized, and their in vitro activities were tested. (Figure 2).

## 2. Results and Discussion

### 2.1. Chemistry

The design of novel antiandrogen compounds was performed to explore several different chemical modifications around the thiohydanthoin and hydantoin scaffolds as depicted in Figure 1, Figure 2 and Figure 3.

Various compounds (±)-**6a-k**, (±)-**7a-j** were synthesized as shown in Figure 1 by deprotonation–hydroxyalkylation of the carbon of the thiohydanthoin ring and the conversion of thiohydantoin to the corresponding hydantoin. The alkylation of commercially available correponding anilines **1a-k** by 2-bromopropionic acid was carried out to obtain compounds **3a-k** in yields ranging from 20% to 88%. Immediate cyclization with 4-isothiocyanato-2-(trifluoromethyl)benzonitrile provided phenylthiohydanthoins **5a-k** as racemic mixtures in good yield (72–76%). The kithiation of **5a-k** with lithium bis(trimethylsilyl)amide at −78 °C was followed by the addition of paraformaldehyde to provide a mixture of diastereoisomeric compounds (±)-**6a-k**, respectively (in 69–81% yield). Then, various thiohydantoins (±)-**6a-k** were transformed to **the** corresponding hydantoins (±)-**7a-j** in excellent yield (76–80%), via sodium periodate and cat. Ruthenium (III) chloride.

As depicted in Figure 2, the reduction of **6h** with Fe/HCl produced compound **8** (91% yield). The compound **8** was acetylated with acetyl chloride to give the final product (±)-**9** in 54% yield. Then, thiohydantoin (±)-**9** was transformed to hydantoin (±)-**10** in 41% yield, via sodium periodate and catalytic ruthenium trichloride. Compounds (±)-**11a,b** were obtained in a high yield (40–78%) from the methoxy derivatives ((±)-**6f**, **6f**) by demethylation using boron tribromide (Figure 2).

Compounds **19a-c** were synthesized according to Figure 3. The key intermediate **17** was generated from the commercially available compound 1 via nitration, bromination, reduction and cyanation (in 60% yield) [13]. Compound **16** reacted with thiophosgene to form the key intermediate thioisocyanate **17** [17]. Cyclizing **3a**, **3c** and **3k** with **17** followed by TEA afforded the compounds **18a-c**. The lithiation of **18a-c** with lithium bis(trimethylsilyl)amide at −78 °C was followed by addition of paraformaldehyde to provide a mixture of diastereoisomeric final products (±)-**19a-c**, respectively (in 46–71% yield).

### 2.2. Antagonist Affinity of Compounds ***6a-l***, ***7a-j***, ***9***, ***10***, ***11a-b*** and ***19a-d***

We examined the AR antagonist activity of the representative compounds **6a-l**, **7a-j**, **9**, **10**, **11a-b** and **19a-d** with a luciferase gene reporter assay in mouse myoblast CV-1 cells.

As shown in Table 1, a series of hydroxymethylthiohydantoin-like SARM compounds exhibited different degrees of AR antagonistic activity. Fortunately, the compounds **6a** and **6c** are comparable to enzalutamide in both their antagonistic activity and efficacy (IC_50_ = 46.8, 46.2, 42.82 nM). What’s more, when the aromatic ring contains a methyl or methoxy group such as **6b** and **6f**, the antagonistic activity is significantly decreased (IC_50_ = 548.5, 286.2 nM), which also indicates that the donor group on the aromatic ring has a negative effect on the antagonistic activity of the compound. A slight decrease in antagonistic activity was also observed when either chlorine or bromine was attached to the aromatic ring (6d, 6e, IC_50_ = 79.27, 116.6 nM). When electron-withdrawing groups such as cyano groups, nitro groups, and ester groups are linked to the aromatic ring, it does not show a positive effect on the antagonistic activity of the compound (**6j**, **6h**, **6j**, IC_50_ = 338.4, 79,616, 531.9 nM). However, the trifluoromethyl-group-containing **6i** (IC_50_ = 68.15 nM) retained the same order of magnitude of antagonistic activity as enzalutamide. No increase in antagonistic activity was observed when **6k**, containing both fluorine and acyl groups on the aromatic ring, was investigated, nor for **6l**, which contains a heterocycloalkoxy group on the aromatic ring (**6k**, **6l**, IC_50_ = 125.75, 337.35 nM). Compoumd **9** and **11a**, containing acyl or hydroxyl groups on the aromatic ring, although inferior to EM, still show a certain antagonistic activity (**9**, **11a**, IC_50_ = 938.8, 3641 nM).

As shown in Table 2, after replacing the sulfur atoms on thiohydantoin with oxygen atoms, although no derivatives were found to be more active than the positive compounds, more compounds showed the same magnitude of antagonistic activity as enzalutamide.

To improve affinity and activity, the methylene group (CH) at the ortho position of the aryl nitrile was replaced with a nitrogen (N) atom, activating the cyano group on the aryl group to form a reversible covalent bond with the endogenous cysteine (Cys784) within the AR ligand-binding pocket. The results in Table 3 show that this strategy was effective. Compound **19a** exhibits excellent antagonistic activity (**19a**, IC_50_ = 18.4 nM), and the antagonistic activity of compound **19b** is almost the same as that of enzalutamide (**19b**, IC_50_ = 32.45 nM). However, the antagonistic activity of **19c** is inferior to that of **6k** (Table 1), which is exactly the opposite of that of **19d**, which better than **6l** (Table 1).

### 2.3. Anti-Proliferative Activity of Compounds ***6a***, ***6c***, ***19a*** and ***19b*** in Prostate Cancer Cell Lines

We performed experiments accordingly to detect the anti-proliferative effect of compounds using the prostate cancer LNCaP cell line. The measurement of proliferative activity was performed after treatment with a concentration range of each compound for 72 h. As can be seen from Table 4, all four compounds exhibited good antiproliferative activity, especially **19b**; its inhibitory effect on the proliferation of LNCAP cell line is even better than that of enzalutamide.

In addition, by using Schrödinger to study the binding mode of compound **19b** to AR (PDB ID: 2OZ7), it was found that the binding posture of **19b** was very similar to that of enzalutamide. The critical interactions of **19b** with the critical residuals are shown in Figure 3a. The cyanyl group of **19b** forms an important hydrogen bond with GLY708, Gln711, and TRP741 via a water molecule, similar to enzalutamide and apalutamide (Figure 3b,c). There is a π–π interaction between the pyridine ring and Phe764, which is similar to apalutamide (Figure 3b). Enzalutamide and apalutamide can form a hydrogen bond with ARG779 due to the presence of an acyl group. Obviously, the absence of an acyl group in **19b** hinders the formation of this hydrogen bond. However, the extra hydroxyl group in **19b** can form a hydrogen bond with ASN705 in the AR, which may be the reason for the high potency of **19b** toward AR.

## 3. Conclusions

In this work, a series of hydroxymethylthiohydantoins, hydroxymethylhydantoins and pyridine SARMs were synthesized, and the remarkable AR antagonistic activities of these compounds were revealed by in vitro cell experiments. Among them, compounds **6a**, **6c**, **6d**, **6i**, **7a-e**, **19a-b,** etc., exhibited the same magnitude of antagonistic activity as enzalutamide. In addition, compounds **6a**, **6c**, **19a** and **19b** exhibited good anti-proliferative activity. The inhibitory effect of **19b** on the proliferation of the LNCAP cell line is even better than that of enzalutamide.

## 4. Experimental Sections

### 4.1. Materials and Methods

All reagents are commercially available and were used without further purification. The the solvents used were of analytical grade. Melting points were taken on a Fisher–Johns melting point apparatus, uncorrected and reported in degrees Centigrade. ^1^H NMR and ^13^C NMR spectra were scanned on a Bruker DRX-400 (400 MHz) using tetramethylsilane (TMS) as an internal standard and using one or two of the following solvents, DMSO-*d6* and CDCl_3_. Chemical shifts are given in δ, ppm. Splitting patterns were designated as follows: s: singlet; d: doublet; t: triplet; q: quartet; m: multiplet. The mass spectra (MS) were recorded on a Finnigan MAT-95 mass spectrometer. The purity of all tested compounds was established by HPLC to be >95.0%. HPLC analysis was performed at room temperature using an Agilent Eclipse XDBC18 (250 mm × 4.6 mm) and as a mobile phase gradient from 5% MeCN/H_2_O (1‰ TFA) for 1 min, 5% MeCN/H_2_O (1‰ TFA) to 95% MeCN/H_2_O (1‰ TFA) for 9 min and 95% MeCN/H_2_O (1‰ TFA) for 5 min more, with a flow rate of 1.0 mL/min and plotted at 254 nm (See Appendix A).

### 4.2. General Synthesis

#### 4.2.1. General Procedure for the Synthesis of **3a-l**

In a 100 mL round-bottomed flask, we added amines (1.0 eq), 2-bromopropanoic acid (1.5 eq) and TEA (3.0 eq) in 150 mL DCM to give a colorless suspension. The reaction mixture was held at room temperature with stirring on for 3 days. The mixture was concentrated by rotovap. One-hundred milliliters water was added. An amount of 60 mL 2M HCl(aq) was added to adjusted pH to 5. The aq layer was extracted with EA. The organic was dried Na_2_SO_4_, filt and conc to give crude product. Then, 20 mL DCM and 50 mL Et_2_O was added. The reaction mixture was filtered through a sintered glass funnel with 50 mL Et_2_O to give **3a-l**.

##### Phenylalanine (**3a**)

White solid (4.5 g, 51% yield). m.p. 157–159 °C; ^1^H NMR (400 MHz, DMSO) δ 7.07 (t, *J* = 7.8 Hz, 2H), 6.61−6.50 (m, 3H), 3.94 (q, *J* = 7.0 Hz, 0H), 1.37 (d, *J* = 7.0 Hz, 3H). ^13^C NMR (101 MHz, DMSO) δ 175.85, 147.65, 128.80, 116.28, 112.41, 50.98, 45.49, 18.11, 8.45.

##### p-tolylalanine (**3b**)

White solid (2.4 g, 29% yield). m.p. 152–154 °C; ^1^H NMR (400 MHz, DMSO) δ 6.88 (d, *J* = 8.2 Hz, 2H), 6.47 (d, *J* = 8.3 Hz, 2H), 3.90 (q, *J* = 7.0 Hz, 1H), 2.14 (s, 3H), 1.35 (d, *J* = 7.0 Hz, 3H). ^13^C NMR (101 MHz, DMSO) δ 175.97, 145.37, 129.24, 124.73, 112.62, 51.27, 20.01, 18.15.

##### (4-fluorophenyl)alanine (**3c**)

White solid (2.7 g, 33% yield). m.p. 150–161 °C; ^1^H NMR (400 MHz, DMSO) δ 6.91 (t, *J* = 8.8 Hz, 2H), 6.54 (dd, *J* = 8.9, 4.5 Hz, 2H), 3.90 (q, *J* = 7.0 Hz, 1H), 1.36 (d, *J* = 6.9 Hz, 3H). ^13^C NMR (101 MHz, DMSO) δ 175.81, 155.60, 153.31, 144.49, 115.26, 115.04, 113.16, 113.09, 51.40, 18.12.

##### (4-chlorophenyl)alanine (**3d**)

White solid (4.2 g, 55% yield). m.p. 143–145 °C; ^1^H NMR (400 MHz, DMSO) δ 7.09 (d, *J* = 8.8 Hz, 2H), 6.55 (d, *J* = 8.8 Hz, 2H), 3.93 (q, *J* = 7.0 Hz, 1H), 1.36 (d, *J* = 7.0 Hz, 3H). ^13^C NMR (101 MHz, DMSO) δ 175.52, 146.70, 128.49, 119.48, 113.73, 50.98, 17.99.

##### (4-bromophenyl)alanine (**3e**)

White solid (5.2 g, 73% yield). m.p. 138–140 °C; ^1^H NMR (400 MHz, DMSO) δ 12.53 (s, 1H), 7.29−7.11 (m, 2H), 6.58−6.45 (m, 2H), 3.92 (q, *J* = 7.0 Hz, 1H), 1.36 (d, *J* = 7.0 Hz, 3H). ^13^C NMR (101 MHz, DMSO) δ 175.49, 147.10, 131.31, 114.28, 106.83, 50.89, 17.98.

##### (4-methoxyphenyl)alanine (**3f**)

White solid (3.8 g, 48% yield). m.p. 150–152 °C; ^1^H NMR (400 MHz, DMSO) δ 8.89 (s, 1H), 6.71 (d, *J* = 8.8 Hz, 2H), 6.52 (d, *J* = 8.8 Hz, 2H), 3.87 (q, *J* = 6.9 Hz, 1H), 3.62 (s, 3H), 1.34 (d, *J* = 6.9 Hz, 3H). ^13^C NMR (101 MHz, DMSO) δ 176.15, 151.02, 141.92, 114.49, 113.53, 55.21, 51.71, 18.26.

##### (4-cyanophenyl)alanine (**3g**)

White solid (5.0 g, 62% yield). m.p. 144–146 °C; ^1^H NMR (400 MHz, DMSO) δ 12.73 (s, 1H), 7.47 (d, *J* = 8.9 Hz, 2H), 6.91 (s, 1H), 6.64 (d, *J* = 8.8 Hz, 2H), 4.07 (d, *J* = 6.7 Hz, 1H), 1.40 (d, *J* = 7.0 Hz, 3H). ^13^C NMR (101 MHz, DMSO) δ 174.79, 151.34, 133.30, 120.39, 112.21, 96.45, 50.40, 17.74.

##### (4-nitrophenyl)alanine (**3h**)

Yellow solid (3.3 g, 44% yield). m.p. 148–150 °C; ^1^H NMR (400 MHz, DMSO) δ 12.83 (s, 1H), 8.01 (d, *J* = 9.0 Hz, 2H), 7.45 (d, *J* = 7.5 Hz, 1H), 6.65 (d, *J* = 9.0 Hz, 2H), 4.17 (p, *J* = 7.0 Hz, 1H), 1.43 (d, *J* = 7.0 Hz, 3H). ^13^C NMR (101 MHz, DMSO) δ 174.35, 153.68, 136.25, 126.04, 111.24, 50.61, 17.68.

##### (4-(trifluoromethyl)phenyl)alanine (**3i**)

White solid (4.3 g, 83% yield). m.p. 197–199 °C; ^1^H NMR (400 MHz, DMSO) δ 12.69 (s, 1H), 7.39 (d, *J* = 8.5 Hz, 2H), 6.66 (d, *J* = 8.5 Hz, 2H), 6.62 (d, *J* = 5.0 Hz, 1H), 4.04 (s, 1H), 1.40 (d, *J* = 7.0 Hz, 3H). ^13^C NMR (101 MHz, DMSO) δ 175.15, 150.84, 129.26, 126.58, 126.18, 126.14, 126.11, 126.07, 123.90, 121.22, 116.30, 115.99, 115.67, 115.35, 111.69, 50.60, 17.85.

##### (4-(methoxycarbonyl)phenyl)alanine (**3j**)

White solid (2.6 g, 88% yield). m.p. 85–87 °C; ^1^H NMR (400 MHz, DMSO) δ 12.69 (s, 1H), 7.38 (d, *J* = 8.5 Hz, 2H), 6.63 (d, *J* = 8.5 Hz, 2H), 6.42 (d, *J* = 5.0 Hz, 1H), 4.04 (q, *J* = 6.9 Hz, 1H), 3.89 (s, 3H), 1.36 (d, *J* = 7.0 Hz, 3H). ^13^C NMR (101 MHz, DMSO) δ 175.15, 165.91, 150.84, 129.26, 126.58, 126.18, 126.14, 126.11, 126.07, 123.90, 121.22, 116.30, 115.99, 115.67, 115.35, 111.69, 50.62, 46.21, 17.75.

##### (3-fluoro-4-(methylcarbamoyl)phenyl)alanine (**3k**)

White solid (2.7 g, 83% yield). m.p. 217–219 °C; ^1^H NMR (400 MHz, DMSO) δ 12.69 (s, 1H), 7.64 (s, 1H), 7.48 (t, *J* = 8.7 Hz, 1H), 6.70 (s, 1H), 6.44 (d, *J* = 8.5 Hz, 1H), 6.31 (d, *J* = 14.6 Hz, 1H), 4.04 (s, 1H), 2.74 (d, *J* = 4.3 Hz, 3H), 1.38 (d, *J* = 6.9 Hz, 3H). ^13^C NMR (101 MHz, DMSO) δ 174.97, 163.84, 162.37, 159.93, 151.86, 151.74, 131.40, 109.76, 109.62, 108.34, 98.11, 97.83, 50.65, 26.21, 17.79.

##### (4-((1-methylpiperidin-4-yl)oxy)phenyl)alanine (**3l**)

White solid (1.6 g, 60% yield). m.p. 104–106 °C; ^1^H NMR (400 MHz, DMSO) δ 6.75 (d, *J* = 8.7 Hz, 2H), 6.50 (d, *J* = 8.7 Hz, 2H), 4.22 (s, 1H), 3.80 (q, *J* = 6.8 Hz, 1H), 2.98 (s, 2H), 2.73 (s, 2H), 2.50 (s, 3H), 1.94 (s, 2H), 1.75 (s, 2H), 1.32 (d, *J* = 6.9 Hz, 3H). ^13^C NMR (101 MHz, DMSO) δ 176.62, 147.76, 142.81, 117.83, 113.35, 70.62, 52.07, 50.74, 43.48, 28.55, 18.45.

#### 4.2.2. General Procedure for the Synthesis of **5a-l** and **18a-d**

In a 100 mL round-bottomed flask, we added **4** or **17** (1.0 eq) and TEA (1.5 eq) in 30 mL CHCl_3_ to give a yellow solution. The reaction vessel was purged with nitrogen. The reaction was heat to 65 °C with stirring on for 1 hr. The reaction mixture was cooled to 25 °C with stirring on. Then, **3a-l** (1.0 eq) was added. The reaction was heated to 65 °C with stirring for 16 h. The mixture was concentrated by rotovap. The crude product was purified by column chromatography to give **5a-l** and **18a-d**.

##### 4-(4-methyl-5-oxo-3-phenyl-2-thioxoimidazolidin-1-yl)-2-(trifluoromethyl)benzonitrile (**5a**)

White solid (1.24 g, 50% yield). m.p. 199–201 °C; ^1^H NMR (400 MHz, CDCl_3_) δ 7.96 (d, *J* = 7.8 Hz, 2H), 7.84 (d, *J* = 8.3 Hz, 1H), 7.53 (t, *J* = 7.5 Hz, 2H), 7.44 (dd, *J* = 12.9, 7.3 Hz, 3H), 4.73 (q, *J* = 7.0 Hz, 1H), 1.55 (d, *J* = 7.1 Hz, 3H). ^13^C NMR (101 MHz, CDCl_3_) δ 179.81, 171.95, 137.17, 136.19, 135.29, 134.05, 133.72, 133.38, 133.05, 132.32, 129.82, 129.63, 129.06, 127.18, 123.27, 122.70, 120.55, 114.88, 110.19, 61.20, 15.94.

##### 4-(4-methyl-5-oxo-2-thioxo-3-(p-tolyl)imidazolidin-1-yl)-2-(trifluoromethyl)benzonitrile (**5b**)

White solid (750 mg, 88% yield). m.p. 192–194 °C; ^1^H NMR (400 MHz, CDCl_3_) δ 7.97 (d, *J* = 7.5 Hz, 2H), 7.84 (dd, *J* = 8.3, 1.6 Hz, 1H), 7.33 (d, *J* = 8.5 Hz, 2H), 7.29 (d, *J* = 8.5 Hz, 2H), 4.69 (q, *J* = 7.0 Hz, 1H), 2.42 (s, 3H), 1.55 (d, *J* = 7.2 Hz, 3H). ^13^C NMR (101 MHz, CDCl_3_) δ 179.87, 172.04, 139.32, 137.17, 135.25, 133.75, 133.52, 133.41, 132.27, 130.47, 127.23, 127.18, 127.13, 127.08, 126.95, 123.25, 120.53, 114.84, 110.20, 61.25, 21.25, 15.93.

##### 4-(3-(4-fluorophenyl)-4-methyl-5-oxo-2-thioxoimidazolidin-1-yl)-2-(trifluoromethyl)benzonitrile (**5c**)

White solid (1.2 g, 58% yield). m.p. 189–191 °C; ^1^H NMR (400 MHz, CDCl_3_) δ 8.04−7.92 (m, 2H), 7.83 (d, *J* = 8.0 Hz, 1H), 7.41 (dd, *J* = 8.7, 4.7 Hz, 2H), 7.22 (t, *J* = 8.4 Hz, 2H), 4.69 (q, *J* = 7.0 Hz, 1H), 1.57 (d, *J* = 7.2 Hz, 4H). ^13^C NMR (101 MHz, CDCl_3_) δ 180.15, 171.71, 163.57, 161.08, 137.02, 135.29, 133.83, 133.50, 132.19, 132.07, 129.29, 129.20, 127.07, 117.07, 116.84, 114.77, 110.33, 61.21, 15.96.

##### 4-(3-(4-chlorophenyl)-4-methyl-5-oxo-2-thioxoimidazolidin-1-yl)-2-(trifluoromethyl)benzonitrile (**5d**)

White solid (1.5 g, 66% yield). m.p. 153–155 °C; ^1^H NMR (400 MHz, CDCl_3_) δ 7.98 (d, *J* = 8.5 Hz, 1H), 7.95 (d, *J* = 1.6 Hz, 1H), 7.82 (dd, *J* = 8.3, 2.0 Hz, 1H), 7.50 (d, *J* = 8.7 Hz, 2H), 7.39 (d, *J* = 8.7 Hz, 2H), 4.72 (q, *J* = 7.1 Hz, 1H), 1.56 (d, *J* = 7.0 Hz, 3H). ^13^C NMR (101 MHz, CDCl_3_) δ 179.84, 171.63, 137.01, 135.31, 134.87, 134.64, 133.81, 132.24, 130.07, 128.47, 127.14, 127.09, 123.22, 120.49, 114.79, 110.32, 60.98, 15.92.

##### 4-(3-(4-bromophenyl)-4-methyl-5-oxo-2-thioxoimidazolidin-1-yl)-2-(trifluoromethyl)benzonitrile (**5e**)

White solid (1.0 g, 50% yield). m.p. 142–144 °C; ^1^H NMR (400 MHz, CDCl_3_) δ 7.98 (d, *J* = 8.3 Hz, 1H), 7.95 (d, *J* = 1.3 Hz, 1H), 7.82 (dd, *J* = 8.3, 1.9 Hz, 1H), 7.66 (d, *J* = 8.3 Hz, 2H), 7.33 (d, *J* = 8.2 Hz, 2H), 4.82−4.65 (m, 1H), 1.56 (d, *J* = 7.0 Hz, 3H). ^13^C NMR (101 MHz, CDCl_3_) δ 179.74, 171.60, 136.97, 135.31, 135.16, 133.84, 133.51, 133.05, 132.22, 128.71, 127.18, 127.13, 127.09, 127.04, 123.21, 122.87, 120.48, 114.77, 110.36, 60.90, 15.94.

##### 4-(3-(4-methoxyphenyl)-4-methyl-5-oxo-2-thioxoimidazolidin-1-yl)-2-(trifluoromethyl)benzonitrile (**5f**)

White solid (1.4 g, 67% yield). m.p. 160–162 °C; ^1^H NMR (400 MHz, CDCl_3_) δ 7.97 (d, *J* = 8.2 Hz, 2H), 7.84 (d, *J* = 9.4 Hz, 1H), 7.32 (d, *J* = 8.9 Hz, 2H), 7.02 (d, *J* = 8.9 Hz, 2H), 4.65 (q, *J* = 7.0 Hz, 1H), 3.86 (s, 3H), 1.55 (d, *J* = 7.0 Hz, 3H). ^13^C NMR (101 MHz, CDCl_3_) δ 180.14, 172.04, 159.81, 137.18, 135.24, 133.75, 133.42, 132.24, 128.66, 128.53, 127.24, 127.16, 127.10, 127.06, 123.25, 120.52, 115.05, 114.83, 110.19, 61.40, 55.56, 15.95.

##### 4-(3-(4-cyanophenyl)-4-methyl-5-oxo-2-thioxoimidazolidin-1-yl)-2-(trifluoromethyl)benzonitrile (**5g**)

White solid (1.7 g, 78% yield). m.p. 169–171 °C; ^1^H NMR (400 MHz, CDCl_3_) δ 7.99 (d, *J* = 8.3 Hz, 1H), 7.95 (d, *J* = 1.6 Hz, 1H), 7.86−7.77 (m, 3H), 7.67 (d, *J* = 8.7 Hz, 2H), 4.87 (q, *J* = 7.0 Hz, 1H), 1.58 (d, *J* = 7.0 Hz, 3H). ^13^C NMR (101 MHz, CDCl_3_) δ 179.54, 171.20, 140.16, 136.87, 135.42, 133.84, 133.53, 132.32, 127.45, 127.22, 127.18, 127.13, 127.08, 123.20, 120.47, 117.81, 114.77, 112.24, 110.46, 60.42, 15.87.

##### 4-(4-methyl-3-(4-nitrophenyl)-5-oxo-2-thioxoimidazolidin-1-yl)-2-(trifluoromethyl)benzonitrile (**5h**)

White solid (1.6 g, 40% yield). m.p. 193–195 °C; ^1^H NMR (400 MHz, CDCl_3_) δ 8.39 (d, *J* = 9.0 Hz, 2H), 8.01 (d, *J* = 8.3 Hz, 1H), 7.95 (d, *J* = 1.4 Hz, 1H), 7.83 (dd, *J* = 8.3, 1.9 Hz, 1H), 7.75 (d, *J* = 9.0 Hz, 2H), 4.92 (q, *J* = 7.0 Hz, 1H), 1.60 (d, *J* = 7.0 Hz, 3H). ^13^C NMR (101 MHz, CDCl_3_) δ 179.52, 171.09, 146.84, 141.72, 136.79, 135.45, 133.94, 133.60, 132.29, 127.38, 127.22, 127.17, 127.12, 127.08, 126.19, 125.01, 123.17, 120.45, 115.65, 114.72, 110.57, 60.43, 15.92.

##### 4-(4-methyl-5-oxo-2-thioxo-3-(4-(trifluoromethyl)phenyl)imidazolidin-1-yl)-2-(trifluoromethyl)benzonitrile (**5i**)

White solid (2.4 g, 82% yield). m.p. 138–140 °C; ^1^H NMR (400 MHz, CDCl_3_) δ 7.99 (d, *J* = 8.3 Hz, 1H), 7.96 (d, *J* = 1.6 Hz, 1H), 7.86−7.76 (m, 3H), 7.63 (d, *J* = 8.3 Hz, 2H), 4.83 (q, *J* = 7.1 Hz, 1H), 1.58 (d, *J* = 7.0 Hz, 3H). ^13^C NMR (101 MHz, CDCl_3_) δ 179.70, 171.41, 139.30, 136.92, 135.37, 134.21, 133.88, 133.54, 133.21, 132.27, 130.92, 130.58, 127.34, 127.21, 127.16, 127.11, 127.07, 126.97, 126.94, 126.90, 126.86, 124.86, 123.20, 122.15, 120.48, 114.76, 110.44, 60.73, 15.92.

##### methyl 4-(3-(4-cyano-3-(trifluoromethyl)phenyl)-5-methyl-4-oxo-2-thioxoimidazolidin-1-yl)benzoate (**5j**)

White solid (400 mg, 32% yield). m.p. 83–85 °C; ^1^H NMR (400 MHz, CDCl_3_) δ 8.20 (d, *J* = 8.3 Hz, 2H), 7.99 (d, *J* = 8.3 Hz, 1H), 7.95 (s, 1H), 7.83 (d, *J* = 8.0 Hz, 1H), 7.58 (d, *J* = 8.3 Hz, 2H), 4.84 (d, *J* = 7.0 Hz, 1H), 3.96 (s, 3H), 1.57 (d, *J* = 6.9 Hz, 3H). 101 MHz, CDCl_3_) δ 179.48, 171.53, 165.90, 140.07, 136.95, 135.84, 135.34, 133.52, 132.28, 131.04, 130.27, 127.18, 127.14, 126.72, 123.20, 121.59, 114.77, 112.01, 110.37, 60.70, 52.50, 15.92.

##### 4-(3-(4-cyano-3-(trifluoromethyl)phenyl)-5-methyl-4-oxo-2-thioxoimidazolidin-1-yl)-2-fluoro-*N*-methylbenzamide (**5k**)

White solid (350 mg, 36% yield). m.p. 205–207 °C; ^1^H NMR (400 MHz, DMSO) δ 8.41 (s, 1H), 8.39 (s, 1H), 8.25 (s, 1H), 8.05 (d, *J* = 8.2 Hz, 1H), 7.77 (t, *J* = 8.0 Hz, 1H), 7.67 (d, *J* = 11.4 Hz, 1H), 7.59 (d, *J* = 8.3 Hz, 1H), 5.28 (q, *J* = 6.7 Hz, 1H), 2.81 (d, *J* = 3.8 Hz, 3H), 1.43 (d, *J* = 6.8 Hz, 3H). ^13^C NMR (101 MHz, DMSO) δ 179.72, 172.19, 163.36, 159.98, 157.51, 139.37, 139.27, 138.07, 136.29, 133.95, 131.30, 130.98, 130.45, 127.89, 127.84, 123.51, 123.39, 123.24, 122.63, 120.79, 114.97, 114.57, 114.31, 108.67, 60.48, 26.23, 14.66.

##### 4-(4-methyl-3-(4-((1-methylpiperidin-4-yl)oxy)phenyl)-5-oxo-2-thioxoimidazolidin-1-yl)-2-(trifluoromethyl)benzonitrile (**5l**)

White solid (465 mg, 42% yield). m.p. 121–123 °C; ^1^H NMR (400 MHz, DMSO) δ 8.38 (d, *J* = 8.3 Hz, 1H), 8.23 (s, 1H), 8.03 (dd, *J* = 8.2, 1.0 Hz, 1H), 7.43 (d, *J* = 8.8 Hz, 2H), 7.08 (d, *J* = 8.9 Hz, 2H), 5.02 (q, *J* = 7.0 Hz, 1H), 4.49−4.33 (m, 1H), 2.67−2.58 (m, 2H), 2.22−2.14 (m, 5H), 1.96 (d, *J* = 10.2 Hz, 2H), 1.71−1.60 (m, 2H), 1.39 (d, *J* = 7.0 Hz, 3H). ^13^C NMR (101 MHz, DMSO) δ 179.75, 172.60, 156.67, 138.28, 136.13, 133.87, 131.17, 130.85, 129.05, 128.63, 128.51, 127.88, 127.84, 127.79, 127.74, 123.55, 120.82, 115.94, 115.01, 108.40, 72.26, 61.07, 52.37, 45.76, 30.55, 14.78.

##### 5-(4-methyl-5-oxo-3-phenyl-2-thioxoimidazolidin-1-yl)-3-(trifluoromethyl)picolinonitrile (**18a**)

White solid (330 mg, 45% yield). m.p. 169–171 °C; ^1^H NMR (400 MHz, DMSO) δ 9.22 (s, 1H), 8.77 (s, 1H), 7.57 (d, *J* = 8.4 Hz, 4H), 7.45 (s, 1H), 5.22 (d, *J* = 4.3 Hz, 1H), 1.42 (d, *J* = 4.1 Hz, 3H). ^13^C NMR (101 MHz, DMSO) δ 179.14, 172.32, 153.66, 136.41, 135.76, 135.69, 135.65, 135.60, 133.45, 129.26, 128.97, 128.86, 128.63, 128.22, 126.97, 122.89, 120.17, 114.21, 60.99, 14.78.

##### 5-(3-(4-fluorophenyl)-4-methyl-5-oxo-2-thioxoimidazolidin-1-yl)-3-(trifluoromethyl)picolinonitrile (**18b**)

White solid (500 mg, 65% yield). m.p. 181–183 °C; ^1^H NMR (400 MHz, CDCl_3_) δ 9.09 (s, 1H), 8.36 (s, 1H), 7.41 (d, *J* = 3.4 Hz, 2H), 7.24 (t, *J* = 8.2 Hz, 2H), 4.74 (d, *J* = 6.9 Hz, 1H), 1.59 (d, *J* = 6.7 Hz, 3H). ^13^C NMR (101 MHz, CDCl_3_) δ 179.36, 171.47, 163.66, 161.17, 152.35, 134.13, 132.33, 131.81, 130.70, 130.36, 130.08, 129.26, 129.17, 122.63, 119.90, 117.18, 116.95, 113.78, 61.35, 15.93.

##### 4-(3-(6-cyano-5-(trifluoromethyl)pyridin-3-yl)-5-methyl-4-oxo-2-thioxoimidazolidin-1-yl)-2-fluoro-*N*-methylbenzamide (**18c**)

White solid (460 mg, 58% yield). m.p. 186–188 °C; ^1^H NMR (400 MHz, DMSO) δ 9.20 (s, 1H), 8.75 (s, 1H), 8.40 (s, 1H), 7.78 (t, *J* = 8.1 Hz, 1H), 7.67 (d, *J* = 11.3 Hz, 1H), 7.59 (d, *J* = 8.3 Hz, 1H), 5.34 (q, *J* = 6.6 Hz, 1H), 2.80 (d, *J* = 4.0 Hz, 3H), 1.44 (d, *J* = 6.8 Hz, 3H). ^13^C NMR (101 MHz, DMSO) δ 179.22, 171.99, 163.32, 160.01, 157.52, 153.66, 139.12, 139.01, 135.77, 135.73, 133.28, 130.53, 129.31, 129.06, 128.70, 123.57, 123.42, 122.86, 122.64, 122.62, 120.13, 114.53, 114.27, 114.20, 79.12, 60.60, 26.23, 14.69.

##### 5-(4-methyl-3-(4-((1-methylpiperidin-4-yl)oxy)phenyl)-5-oxo-2-thioxoimidazolidin-1-yl)-3-(trifluoromethyl)picolinonitrile (**18d**)

White solid (600 mg, 70% yield). m.p. 112–114 °C; ^1^H NMR (400 MHz, CDCl_3_) δ 9.09 (s, 1H), 8.36 (s, 1H), 7.30 (d, *J* = 8.4 Hz, 2H), 7.02 (d, *J* = 8.5 Hz, 2H), 4.69 (q, *J* = 7.0 Hz, 1H), 4.37 (s, 1H), 2.71 (s, 2H), 2.32 (s, 3H), 2.30 (s, 2H), 2.08−2.00 (m, 2H), 1.88 (dd, *J* = 14.6, 5.8 Hz, 2H), 1.58 (d, *J* = 7.0 Hz, 3H). ^13^C NMR (101 MHz, CDCl_3_) δ 179.25, 171.76, 157.96, 152.35, 134.11, 134.07, 132.43, 130.64, 130.30, 129.98, 128.42, 128.16, 122.66, 119.93, 118.42, 116.70, 113.89, 113.77, 72.55, 61.49, 52.61, 46.20, 30.80, 15.95.

#### 4.2.3. General Procedure for the Synthesis of **6a-l** and **19a-d**

In a 50 mL round-bottomed flask, we added compound **5a-l** or **18a-d** (1.0 eq) in 10 mL THF to give a colorless solution. The reaction vessel was purged with nitrogen. The reaction mixture was cooled to −78 °C with stirring on. LiHMDS (1.3 eq) was added. The reaction mixture was held at −78 °C with stirring for 10 min. Formaldehyde (3.0 eq) was added. The reaction mixture was warmed up to rt with stirring on for 30 min. Then, 10 mL sat. NH_4_Cl (aq) was added. The aq layer was extracted with EA. We combined the organic layers and washed them with brine. The organic was dried with Na_2_SO_4_, filt and conc to give crude product. The crude product was purified by column chromatography to give **6a-l** and **19a-d**.

##### 4-(4-(hydroxymethyl)-4-methyl-5-oxo-3-phenyl-2-thioxoimidazolidin-1-yl)-2-(trifluoromethyl)benzonitrile (**6a**)

White solid (45 mg, 21% yield). m.p. 216–219 °C; ^1^H NMR (400 MHz, DMSO) δ 8.39 (d, *J* = 8.2 Hz, 1H), 8.15 (d, *J* = 1.6 Hz, 1H), 7.99 (dd, *J* = 8.2, 1.8 Hz, 1H), 7.54 (dq, *J* = 14.5, 7.2 Hz, 3H), 7.48−7.41 (m, 2H), 5.87 (t, *J* = 5.1 Hz, 1H), 3.78 (dd, *J* = 11.7, 4.7 Hz, 1H), 3.40 (dd, *J* = 11.7, 5.5 Hz, 1H), 1.35 (s, 3H). ^13^C NMR (101 MHz, DMSO) δ 180.92, 173.96, 137.95, 136.26, 135.22, 133.68, 131.27, 130.95, 129.63, 129.47, 129.22, 127.51, 127.47, 127.42, 127.38, 123.50, 120.78, 114.96, 108.51, 108.49, 71.63, 63.62, 16.78. HRMS(ESI)*m/z* calcd for C_19_H_14_F_3_N_3_O_2_S [M+H]^+^: 406.0837, Found: 406.0871.

##### 4-(4-(hydroxymethyl)-4-methyl-5-oxo-2-thioxo-3-(p-tolyl)imidazolidin-1-yl)-2-(trifluoromethyl)benzonitrile (**6b**)

White solid (70 mg, 43% yield). m.p. 190–192 °C; ^1^H NMR (400 MHz, DMSO) δ 8.39 (d, *J* = 8.2 Hz, 1H), 8.14 (s, 1H), 7.98 (d, *J* = 8.2 Hz, 1H), 7.36 (d, *J* = 8.2 Hz, 2H), 7.32 (d, *J* = 8.3 Hz, 2H), 5.83 (t, *J* = 5.1 Hz, 1H), 3.77 (dd, *J* = 11.6, 4.8 Hz, 1H), 3.39 (dd, *J* = 11.6, 5.6 Hz, 1H), 2.38 (s, 3H), 1.33 (s, 3H). ^13^C NMR (101 MHz, DMSO) δ 180.95, 174.01, 138.81, 137.99, 136.24, 133.67, 132.55, 131.25, 131.09, 130.92, 129.97, 129.32, 127.46, 127.42, 120.77, 114.96, 108.47, 71.55, 63.58, 20.74, 16.75. HRMS(ESI)*m/z* calcd for C_20_H_16_F_3_N_3_O_2_S [M+H]^+^: 420.0994, Found: 420.0973.

##### 4-(3-(4-fluorophenyl)-4-(hydroxymethyl)-4-methyl-5-oxo-2-thioxoimidazolidin-1-yl)-2-(trifluoromethyl)benzonitrile (**6c**)

White solid (90 mg, 42% yield). m.p. 231–233 °C; ^1^H NMR (400 MHz, DMSO) δ 8.39 (d, *J* = 8.2 Hz, 1H), 8.14 (d, *J* = 0.9 Hz, 1H), 7.99 (dd, *J* = 8.2, 1.3 Hz, 1H), 7.50 (dd, *J* = 8.8, 5.1 Hz, 2H), 7.42 (t, *J* = 8.8 Hz, 2H), 5.89 (t, *J* = 5.2 Hz, 1H), 3.79 (dd, *J* = 11.7, 4.9 Hz, 1H), 3.40 (dd, *J* = 11.7, 5.7 Hz, 1H), 1.36 (s, 3H). ^13^C NMR (101 MHz, DMSO) δ 181.25, 173.90, 163.29, 160.84, 137.88, 136.29, 133.63, 131.96, 131.87, 131.42, 131.39, 131.31, 130.98, 127.41, 127.36, 123.48, 120.76, 116.58, 116.34, 114.94, 108.57, 108.55, 71.62, 63.58, 16.65. HRMS(ESI)*m/z* calcd for C_19_H_13_F_4_N_3_O_2_S [M+H]^+^: 424.0743, Found: 424.0715.

##### 4-(3-(4-chlorophenyl)-4-(hydroxymethyl)-4-methyl-5-oxo-2-thioxoimidazolidin-1-yl)-2-(trifluoromethyl)benzonitrile (**6d**)

White solid (73 mg, 34% yield). m.p. 205–207 °C; ^1^H NMR (400 MHz, DMSO) δ 8.40 (d, *J* = 8.2 Hz, 1H), 8.15 (s, 1H), 7.99 (d, *J* = 8.2 Hz, 1H), 7.66 (d, *J* = 8.5 Hz, 2H), 7.48 (d, *J* = 8.5 Hz, 2H), 5.90 (t, *J* = 4.9 Hz, 1H), 3.80 (dd, *J* = 11.7, 4.6 Hz, 1H), 3.42 (dd, *J* = 11.7, 5.6 Hz, 1H), 1.37 (s, 3H). ^13^C NMR (101 MHz, DMSO) δ 181.10, 173.86, 137.83, 136.31, 134.13, 133.95, 133.63, 131.59, 131.31, 130.99, 129.65, 127.41, 127.36, 123.48, 120.76, 114.93, 108.60, 71.71, 63.60, 16.66. HRMS(ESI)*m/z* calcd for C_19_H_13_ClF_3_N_3_O_2_S [M+H]^+^: 440.0447, Found: 440.0476.

##### 4-(3-(4-bromophenyl)-4-(hydroxymethyl)-4-methyl-5-oxo-2-thioxoimidazolidin-1-yl)-2-(trifluoromethyl)benzonitrile (**6e**)

White solid (40 mg, 19% yield). m.p. 206–208 °C; ^1^H NMR (400 MHz, DMSO) δ 8.40 (d, *J* = 8.3 Hz, 1H), 8.14 (d, *J* = 1.5 Hz, 1H), 7.98 (dd, *J* = 8.2, 1.7 Hz, 1H), 7.79 (d, *J* = 8.6 Hz, 2H), 7.41 (d, *J* = 8.6 Hz, 2H), 5.89 (t, *J* = 5.2 Hz, 1H), 3.79 (dd, *J* = 11.8, 4.7 Hz, 1H), 3.41 (dd, *J* = 11.8, 5.6 Hz, 1H), 1.36 (s, 3H). ^13^C NMR (101 MHz, DMSO) δ 181.00, 173.89, 137.79, 136.29, 134.51, 133.64, 132.62, 131.87, 131.33, 131.01, 127.43, 127.38, 123.45, 122.61, 120.72, 114.94, 108.59, 71.69, 63.54, 16.64. HRMS(ESI)*m/z* calcd for C_19_H_13_BrF_3_N_3_O_2_S [M+H]^+^: 483.9942, Found: 483.9978.

##### 4-(4-(hydroxymethyl)-3-(4-methoxyphenyl)-4-methyl-5-oxo-2-thioxoimidazolidin-1-yl)-2-(trifluoromethyl)benzonitrile (**6f**)

White solid (200 mg, 93% yield). m.p. 183–185 °C; ^1^H NMR (400 MHz, DMSO) δ 8.39 (d, *J* = 8.3 Hz, 1H), 8.14 (d, *J* = 1.6 Hz, 1H), 7.99 (dd, *J* = 8.2, 1.7 Hz, 1H), 7.37 (d, *J* = 8.9 Hz, 2H), 7.10 (d, *J* = 8.9 Hz, 2H), 5.84 (t, *J* = 5.2 Hz, 1H), 3.83 (s, 3H), 3.78 (dd, *J* = 11.7, 4.8 Hz, 1H), 3.40 (dd, *J* = 11.7, 5.7 Hz, 1H), 1.34 (s, 3H). ^13^C NMR (101 MHz, DMSO) δ 181.17, 174.04, 159.50, 138.02, 136.23, 133.65, 130.73, 127.54, 127.45, 127.40, 123.50, 120.78, 114.96, 114.62, 108.45, 71.52, 63.56, 55.35, 16.71. HRMS(ESI)*m/z* calcd for C_20_H_16_F_3_N_3_O_3_S [M+H]^+^: 436.0943, Found: 436.0911.

##### 4-(3-(4-cyanophenyl)-4-(hydroxymethyl)-4-methyl-5-oxo-2-thioxoimidazolidin-1-yl)-2-(trifluoromethyl)benzonitrile (**6g**)

White solid (130 mg, 60% yield). m.p. 234–236 °C; ^1^H NMR (400 MHz, DMSO) δ 8.40 (d, *J* = 8.2 Hz, 1H), 8.14 (s, 1H), 8.09 (d, *J* = 8.5 Hz, 2H), 7.99 (d, *J* = 8.2 Hz, 1H), 7.67 (d, *J* = 8.5 Hz, 2H), 5.94 (t, *J* = 5.3 Hz, 1H), 3.81 (dd, *J* = 11.9, 4.9 Hz, 1H), 3.42 (dd, *J* = 11.9, 5.8 Hz, 1H), 1.38 (s, 3H). ^13^C NMR (101 MHz, DMSO) δ 181.00, 173.70, 139.63, 139.13, 137.69, 136.36, 133.73, 133.63, 131.04, 127.39, 127.34, 124.87, 118.11, 114.92, 112.11, 108.71, 72.03, 63.70, 16.68. HRMS(ESI)*m/z* calcd for C_20_H_13_F_3_N_4_O_2_S [M+H]^+^: 431.0790, Found: 431.0771.

##### 4-(4-(hydroxymethyl)-4-methyl-3-(4-nitrophenyl)-5-oxo-2-thioxoimidazolidin-1-yl)-2-(trifluoromethyl)benzonitrile (**6h**)

Yellow solid (330 mg, 39% yield). m.p. 206–208 °C; ^1^H NMR (400 MHz, DMSO) δ 8.45 (d, *J* = 8.9 Hz, 2H), 8.42 (d, *J* = 8.4 Hz, 1H), 8.17 (s, 1H), 8.02 (d, *J* = 8.3 Hz, 1H), 7.77 (d, *J* = 9.0 Hz, 2H), 5.98 (t, *J* = 5.3 Hz, 1H), 3.85 (dd, *J* = 11.9, 5.0 Hz, 1H), 3.45 (dd, *J* = 11.9, 5.8 Hz, 1H), 1.41 (s, 3H). ^13^C NMR (101 MHz, DMSO) δ 181.06, 173.69, 147.57, 141.24, 137.69, 136.38, 133.65, 131.37, 131.06, 127.46, 127.41, 127.37, 127.32, 126.19, 124.85, 123.47, 120.74, 114.91, 108.74, 108.73, 72.13, 63.71, 16.70. HRMS(ESI)*m/z* calcd for C_19_H_13_F_3_N_4_O_4_S [M+H]^+^: 451.688, Found: 451.0651.

##### 4-(4-(hydroxymethyl)-4-methyl-5-oxo-2-thioxo-3-(4-(trifluoromethyl)phenyl)imidazolidin-1-yl)-2-(trifluoromethyl)benzonitrile (**6i**)

White solid (800 mg, 75% yield). m.p. 178–180 °C; ^1^H NMR (400 MHz, DMSO) δ 8.41 (d, *J* = 8.2 Hz, 1H), 8.16 (s, 1H), 7.99 (t, *J* = 7.8 Hz, 3H), 7.70 (d, *J* = 8.1 Hz, 2H), 5.94 (t, *J* = 5.2 Hz, 1H), 3.82 (dd, *J* = 11.8, 4.9 Hz, 1H), 3.43 (dd, *J* = 11.8, 5.7 Hz, 1H), 1.39 (s, 3H). ^13^C NMR (101 MHz, DMSO) δ 181.11, 173.78, 139.09, 137.77, 136.32, 133.65, 131.37, 131.04, 130.86, 130.03, 129.72, 129.40, 129.08, 127.46, 127.42, 127.37, 127.32, 126.71, 126.68, 125.20, 123.47, 123.35, 122.50, 120.75, 119.79, 118.51, 114.91, 108.69, 108.67, 71.93, 63.67, 16.68. HRMS(ESI)*m/z* calcd for C_20_H_13_F_6_N_3_O_2_S [M+Na]^+^: 493.0530, Found: 496.0558.

##### methyl 4-(3-(4-cyano-3-(trifluoromethyl)phenyl)-5-(hydroxymethyl)-5-methyl-4-oxo-2-thioxoimidazolidin-1-yl)benzoate (**6j**)

White solid (110 mg, 41% yield). m.p. 168–170 °C; ^1^H NMR (400 MHz, DMSO) δ 8.40 (d, *J* = 8.3 Hz, 1H), 8.15 (d, *J* = 8.5 Hz, 3H), 8.00 (dd, *J* = 8.2, 1.6 Hz, 1H), 7.62 (d, *J* = 8.5 Hz, 2H), 5.92 (t, *J* = 5.3 Hz, 1H), 3.90 (s, 3H), 3.81 (dd, *J* = 11.8, 4.9 Hz, 1H), 3.42 (dd, *J* = 11.8, 5.7 Hz, 1H), 1.38 (s, 3H). ^13^C NMR (101 MHz, DMSO) δ 180.90, 173.80, 165.54, 139.58, 137.80, 136.45, 136.32, 133.66, 131.33, 131.00, 130.37, 130.26, 130.22, 127.45, 127.40, 123.48, 121.63, 120.76, 114.92, 108.64, 71.91, 63.69, 52.38, 16.75. HRMS(ESI)*m/z* calcd for C_21_H_16_F_3_N_3_O_4_S [M+H]^+^: 464.0892, Found: 464.0911.

##### 4-(3-(4-cyano-3-(trifluoromethyl)phenyl)-5-(hydroxymethyl)-5-methyl-4-oxo-2-thioxoimidazolidin-1-yl)-2-fluoro-*N*-methylbenzamide (**6k**)

White solid (60 mg, 56% yield). m.p. 123–125 °C; ^1^H NMR (400 MHz, CDCl_3_) δ 8.09 (t, *J* = 8.2 Hz, 1H), 7.92 (d, *J* = 9.4 Hz, 2H), 7.79 (d, *J* = 8.2 Hz, 1H), 7.34 (d, *J* = 8.3 Hz, 1H), 7.29 (d, *J* = 11.8 Hz, 1H), 6.86 (dd, *J* = 9.8, 4.6 Hz, 1H), 3.98 (d, *J* = 11.3 Hz, 1H), 3.69 (s, 1H), 3.53 (d, *J* = 11.4 Hz, 1H), 2.99 (d, *J* = 4.1 Hz, 3H), 1.42 (s, 3H). ^13^C NMR (101 MHz, CDCl_3_) δ 181.45, 173.70, 163.43, 161.65, 159.16, 139.27, 139.16, 137.28, 135.37, 133.67, 133.34, 132.85, 132.47, 127.26, 127.21, 126.45, 123.22, 122.52, 122.39, 120.50, 118.35, 118.08, 114.86, 110.09, 71.68, 64.19, 27.05, 17.88. HRMS(ESI)*m/z* calcd for C_21_H_16_F_4_N_4_O_3_S [M+H]^+^: 491.0958, Found: 481.0950.

##### 4-(4-(hydroxymethyl)-4-methyl-3-(4-((1-methylpiperidin-4-yl)oxy)phenyl)-5-oxo-2-thioxoimidazolidin-1-yl)-2-(trifluoromethyl)benzonitrile (**6l**)

White solid (40 mg, 38% yield). m.p. 117–119 °C; ^1^H NMR (400 MHz, CDCl_3_) δ 7.95 (d, *J* = 7.9 Hz, 2H), 7.85 (d, *J* = 8.2 Hz, 1H), 7.35 (d, *J* = 8.8 Hz, 2H), 7.00 (d, *J* = 8.9 Hz, 2H), 4.42 (s, 1H), 3.96 (d, *J* = 11.2 Hz, 1H), 3.78 (s, 1H), 3.59 (d, *J* = 11.2 Hz, 1H), 2.86−2.76 (m, 2H), 2.55 (s, 2H), 2.41 (s, 3H), 2.11 (s, 2H), 1.93 (s, 2H), 1.43 (s, 3H). ^13^C NMR (101 MHz, CDCl_3_) δ 181.74, 174.32, 157.98, 137.58, 135.26, 133.56, 133.23, 132.53, 130.84, 127.46, 127.30, 127.25, 123.29, 120.56, 116.54, 114.91, 109.92, 77.28, 71.45, 63.99, 51.88, 45.50, 29.73, 17.85. HRMS(ESI)*m/z* calcd for C_25_H_25_F_3_N_4_O_3_S [M+H]^+^: 519.1678, Found: 519.1692.

##### 5-(4-(hydroxymethyl)-4-methyl-5-oxo-3-phenyl-2-thioxoimidazolidin-1-yl)-3-(trifluoromethyl)picolinonitrile (**19a**)

White solid (50 mg, 31% yield). m.p. 203–205 °C; ^1^H NMR (400 MHz, DMSO) δ 9.18 (d, *J* = 1.3 Hz, 1H), 8.71 (d, *J* = 1.4 Hz, 1H), 7.61−7.52 (m, 3H), 7.46 (d, *J* = 7.4 Hz, 2H), 5.90 (t, *J* = 5.3 Hz, 1H), 3.82 (dd, *J* = 11.8, 4.6 Hz, 1H), 3.41 (dd, *J* = 11.8, 6.0 Hz, 1H), 1.37 (s, 3H). ^13^C NMR (101 MHz, DMSO) δ 180.36, 173.77, 153.28, 135.28, 135.25, 134.98, 133.12, 129.56, 129.52, 129.38, 128.85, 128.67, 128.34, 114.21, 71.95, 63.60, 16.79. HRMS(ESI) *m/z* calcd for C_18_H_13_F_3_N_4_O_2_S [M+H]^+^: 407.0790, Found: 407.0789.

##### 5-(3-(4-fluorophenyl)-4-(hydroxymethyl)-4-methyl-5-oxo-2-thioxoimidazolidin-1-yl)-3-(trifluoromethyl)picolinonitrile (**19b**)

White solid (130 mg, 60% yield). m.p. 181–183 °C; ^1^H NMR (400 MHz, DMSO) δ 9.21 (s, 1H), 8.73 (s, 1H), 7.53 (s, 2H), 7.46 (s, 2H), 5.96 (s, 1H), 3.87 (d, *J* = 7.7 Hz, 1H), 3.45 (d, *J* = 5.5 Hz, 1H), 1.41 (s, 3H). ^13^C NMR (101 MHz, DMSO) δ 180.68, 173.71, 163.35, 160.90, 153.23, 135.23, 135.19, 133.06, 131.87, 131.77, 131.18, 131.15, 129.03, 128.91, 122.83, 120.11, 116.67, 116.44, 114.18, 71.94, 63.55, 16.65. HRMS(ESI)*m/z* calcd for C_18_H_12_F_4_N_4_O_2_S [M+H]^+^: 425.0695, Found: 425.0699.

##### 4-(3-(6-cyano-5-(trifluoromethyl)pyridin-3-yl)-5-(hydroxymethyl)-5-methyl-4-oxo-2-thioxoimidazolidin-1-yl)-2-fluoro-*N*-methylbenzamide (**19c**)

White solid (60 mg, 56% yield). m.p. 127–129 °C; ^1^H NMR (400 MHz, CDCl_3_) δ 9.01 (s, 1H), 8.32 (s, 1H), 8.12 (t, *J* = 7.9 Hz, 1H), 7.36 (d, *J* = 8.0 Hz, 1H), 7.30 (d, *J* = 11.5 Hz, 1H), 6.87 (s, 1H), 4.16 (s, 1H), 4.02 (d, *J* = 11.2 Hz, 1H), 3.58 (d, *J* = 11.2 Hz, 1H), 3.01 (s, 3H), 1.46 (s, 3H). ^13^C NMR (101 MHz, CDCl_3_) δ 180.68, 173.43, 163.40, 161.67, 159.17, 152.51, 139.00, 138.88, 134.37, 132.96, 132.57, 130.69, 130.34, 129.93, 126.38, 122.60, 122.46, 119.88, 118.30, 118.03, 113.80, 71.92, 64.21, 27.10, 17.84. HRMS(ESI)*m/z* calcd for C_20_H_15_F_4_N_5_O_3_S [M+H]^+^: 482.0910, Found: 482.0939.

##### 5-(4-(hydroxymethyl)-4-methyl-3-(4-((1-methylpiperidin-4-yl)oxy)phenyl)-5-oxo-2-thioxoimidazolidin-1-yl)-3-(trifluoromethyl)picolinonitrile (**19d**)

White solid (38 mg, 36% yield). m.p. 119–121 °C; ^1^H NMR (400 MHz, CDCl_3_) δ 9.01 (s, 1H), 8.29 (s, 1H), 7.27 (d, *J* = 8.5 Hz, 2H), 6.94 (d, *J* = 8.7 Hz, 2H), 4.33 (s, 1H), 4.00 (s, 1H), 3.90 (d, *J* = 11.0 Hz, 1H), 3.54 (d, *J* = 11.0 Hz, 1H), 2.70 (s, 2H), 2.37 (s, 2H), 2.29 (s, 3H), 2.00 (s, 2H), 1.83 (s, 2H), 1.38 (s, 3H). ^13^C NMR (101 MHz, CDCl_3_) δ 179.95, 173.09, 157.20, 151.55, 133.28, 133.24, 131.82, 129.67, 129.57, 129.23, 128.75, 126.03, 121.67, 118.94, 115.58, 112.83, 76.24, 70.67, 62.93, 51.14, 44.75, 29.09, 16.80. HRMS(ESI)*m/z* calcd for C_24_H_24_F_3_N_5_O_3_S [M+H]^+^: 520.1630, Found: 520.1649.

#### 4.2.4. General Procedure for the Synthesis of **7a-j** and **10**

In a 25 mL round-bottomed flask, we added **6a-j** and **9** (1.0 eq) in 1 mL CCl_4_ and 1 mL MeCN to give a colorless solution. The reaction mixture was cooled to 0 °C with stirring on. NaIO_4_ (4.0 eq) in 2 mL water was added. Ruthenium (III) chloride (0.05 eq) was added. The reaction mixture was held at rt with stirring on for 2 h. Then, 2 mL NaHCO_3_ (aq) was added. The aq layer was extracted with DCM. We combined the organic layers and washed them with brine. The organic was dried with Na_2_SO_4_, filt and conc to give a crude product. The crude product was purified by column chromatography to give **7a-j** and **11**.

##### 4-(4-(hydroxymethyl)-4-methyl-2,5-dioxo-3-phenylimidazolidin-1-yl)-2-(trifluoromethyl)benzonitrile (**7a**)

White solid (40 mg, 48% yield). m.p. 187–189 °C; ^1^H NMR (400 MHz, DMSO) δ 8.36 (d, *J* = 8.3 Hz, 1H), 8.20 (s, 1H), 8.08 (d, *J* = 8.0 Hz, 1H), 7.56−7.43 (m, 5H), 5.72 (t, *J* = 4.7 Hz, 1H), 3.76 (dd, *J* = 11.5, 4.7 Hz, 1H), 3.38 (dd, *J* = 11.5, 5.2 Hz, 1H), 1.32 (s, 3H). ^13^C NMR (101 MHz, DMSO) δ 173.20, 153.19, 136.59, 136.34, 133.48, 131.73, 131.42, 131.10, 130.78, 129.80, 129.35, 128.63, 123.69, 123.64, 120.82, 115.12, 106.96, 68.41, 63.38, 17.09. HRMS(ESI)*m/z* calcd for C_19_H_14_F_3_N_3_O_3_ [M+H]^+^: 390.1066, Found: 390.1074.

##### 4-(4-(hydroxymethyl)-4-methyl-2,5-dioxo-3-(p-tolyl)imidazolidin-1-yl)-2-(trifluoromethyl)benzonitrile (**7b**)

White solid (20 mg, 49% yield). m.p. 156–158 °C; ^1^H NMR (400 MHz, DMSO) δ 8.35 (d, *J* = 8.4 Hz, 1H), 8.18 (d, *J* = 1.6 Hz, 1H), 8.06 (dd, *J* = 8.4, 1.7 Hz, 1H), 7.34 (d, *J* = 9.0 Hz, 4H), 5.68 (t, *J* = 5.1 Hz, 1H), 3.73 (dd, *J* = 11.6, 4.8 Hz, 1H), 3.35 (dd, *J* = 11.6, 4.8 Hz, 1H), 2.37 (s, 3H), 1.29 (s, 3H). ^13^C NMR (101 MHz, DMSO) δ 173.25, 153.20, 138.23, 136.64, 136.32, 131.41, 131.09, 130.70, 129.84, 129.77, 129.21, 123.66, 123.61, 123.54, 120.82, 115.12, 106.92, 106.90, 68.29, 63.31, 20.67, 17.05. HRMS(ESI)*m/z* calcd for C_20_H_16_F_3_N_3_O_3_ [M+H]^+^: 404.1222, Found: 404.1219.

##### 4-(3-(4-fluorophenyl)-4-(hydroxymethyl)-4-methyl-2,5-dioxoimidazolidin-1-yl)-2-(trifluoromethyl)benzonitrile (**7c**)

White solid (50 mg, 43% yield). m.p. 208–210 °C; ^1^H NMR (400 MHz, DMSO) δ 8.36 (d, *J* = 8.4 Hz, 1H), 8.18 (s, 1H), 8.07 (d, *J* = 8.4 Hz, 1H), 7.50 (dd, *J* = 8.8, 5.1 Hz, 2H), 7.38 (t, *J* = 8.8 Hz, 2H), 5.73 (t, *J* = 5.2 Hz, 1H), 3.75 (dd, *J* = 11.7, 5.1 Hz, 1H), 3.36 (dd, *J* = 11.7, 5.4 Hz, 1H), 1.31 (s, 3H).^13^C NMR (101 MHz, DMSO) δ 173.13, 162.99, 160.55, 153.28, 136.53, 136.37, 131.70, 131.61, 131.44, 131.11, 129.79, 129.60, 123.67, 123.62, 120.81, 116.40, 116.18, 115.11, 107.00, 68.39, 63.27, 16.91. HRMS(ESI)*m/z* calcd for C_19_H_13_F_4_N_3_O_3_ [M+H]^+^: 408.0971, Found: 408.0941.

##### 4-(3-(4-chlorophenyl)-4-(hydroxymethyl)-4-methyl-2,5-dioxoimidazolidin-1-yl)-2-(trifluoromethyl)benzonitrile (**7d**)

White solid (20 mg, 41% yield). m.p. 152–154 °C; ^1^H NMR (400 MHz, DMSO) δ 8.36 (d, *J* = 8.4 Hz, 1H), 8.18 (d, *J* = 1.4 Hz, 1H), 8.06 (dd, *J* = 8.4, 1.8 Hz, 1H), 7.61 (d, *J* = 8.7 Hz, 2H), 7.48 (d, *J* = 8.6 Hz, 2H), 5.73 (t, *J* = 5.2 Hz, 1H), 3.76 (dd, *J* = 11.8, 5.1 Hz, 1H), 3.39 (dd, *J* = 11.8, 5.4 Hz, 1H), 1.32 (s, 3H). ^13^C NMR (101 MHz, DMSO) δ 173.07, 153.17, 136.46, 136.37, 136.32, 133.19, 132.47, 131.44, 131.07, 130.79, 129.86, 129.46, 123.72, 123.67, 123.52, 120.80, 118.07, 117.30, 115.54, 115.10, 110.74, 107.06, 68.53, 63.36, 40.11, 39.90, 39.70, 39.49, 39.28, 39.07, 38.86, 16.96. HRMS(ESI)*m/z* calcd for C_19_H_13_ClF_3_N_3_O_3_ [M+H]^+^: 424.0676, Found: 424.0640.

##### 4-(3-(4-bromophenyl)-4-(hydroxymethyl)-4-methyl-2,5-dioxoimidazolidin-1-yl)-2-(trifluoromethyl)benzonitrile (**7e**)

White solid (50 mg, 19% yield). m.p. 102–104 °C; ^1^H NMR (400 MHz, DMSO) δ 10.71 (s, 1H), 8.29 (s, 1H), 8.18 (dd, *J* = 8.6, 1.5 Hz, 1H), 8.14 (d, *J* = 8.7 Hz, 1H), 7.60 (d, *J* = 8.8 Hz, 2H), 7.31 (d, *J* = 8.8 Hz, 2H), 4.74 (d, *J* = 9.2 Hz, 1H), 4.37 (d, *J* = 9.2 Hz, 1H), 1.64 (s, 3H). ^13^C NMR (101 MHz, DMSO) δ 173.07, 153.12, 136.46, 136.36, 136.31, 132.94, 132.41, 131.45, 131.35, 131.12, 129.85, 123.77, 123.72, 123.67, 123.62, 123.52, 121.72, 120.79, 115.55, 115.09, 107.08, 68.51, 63.37, 16.97. HRMS(ESI)*m/z* calcd for C_19_H_13_BrF_3_N_3_O_3_ [M+H]^+^: 468.0171, Found: 468.0135.

##### 4-(4-(hydroxymethyl)-3-(4-methoxyphenyl)-4-methyl-2,5-dioxoimidazolidin-1-yl)-2-(trifluoromethyl)benzonitrile (**7f**)

White solid (160 mg, 74% yield). m.p. 128–130 °C; ^1^H NMR (400 MHz, DMSO) δ 8.35 (d, *J* = 5.6 Hz, 1H), 8.18 (s, 1H), 8.07 (d, *J* = 5.8 Hz, 1H), 7.37 (d, *J* = 5.0 Hz, 2H), 7.06 (d, *J* = 5.8 Hz, 2H), 5.69 (s, 1H), 3.81 (s, 3H), 3.73 (s, 1H), 3.33 (s, 1H), 1.29 (s, 3H). ^13^C NMR (101 MHz, DMSO) δ 173.28, 159.26, 153.33, 136.66, 136.32, 130.78, 129.73, 125.59, 123.65, 123.62, 123.57, 123.53, 120.81, 120.74, 120.65, 115.13, 114.54, 113.98, 106.86, 68.21, 63.22, 55.34, 16.95. HRMS(ESI)*m/z* calcd for C_20_H_16_F_3_N_3_O_4_ [M+H]^+^: 420.1171, Found: 420.1199.

##### 4-(3-(4-cyanophenyl)-4-(hydroxymethyl)-4-methyl-2,5-dioxoimidazolidin-1-yl)-2-(trifluoromethyl)benzonitrile (**7g**)

White solid (22 mg, 46% yield). m.p. 169–171 °C; ^1^H NMR (400 MHz, DMSO) δ 8.43 (d, *J* = 8.4 Hz, 1H), 8.25 (d, *J* = 1.3 Hz, 1H), 8.12 (dd, *J* = 8.4, 1.5 Hz, 1H), 8.07 (d, *J* = 8.6 Hz, 2H), 7.75 (d, *J* = 8.6 Hz, 2H), 5.83 (t, *J* = 5.3 Hz, 1H), 3.85 (dd, *J* = 11.9, 5.2 Hz, 1H), 3.56 (dd, *J* = 11.9, 5.5 Hz, 2H), 1.44 (s, 3H). ^13^C NMR (101 MHz, DMSO) δ 172.88, 153.06, 138.49, 136.43, 136.27, 133.49, 131.48, 131.15, 130.83, 130.03, 129.12, 123.89, 123.84, 123.51, 120.78, 118.27, 115.08, 110.66, 107.29, 68.98, 63.64, 17.11. HRMS(ESI)*m/z* calcd for C_20_H_13_F_3_N_4_O_3_ [M+H]^+^: 415.1018, Found: 415.1016.

##### 4-(4-(hydroxymethyl)-4-methyl-3-(4-nitrophenyl)-2,5-dioxoimidazolidin-1-yl)-2-(trifluoromethyl)benzonitrile (**7h**)

Light-yellow solid (60 mg, 48% yield). m.p. 160–162 °C; ^1^H NMR (400 MHz, DMSO) δ 8.50−8.30 (m, 3H), 8.20 (d, *J* = 1.0 Hz, 1H), 8.07 (dd, *J* = 8.4, 1.4 Hz, 1H), 7.78 (d, *J* = 8.9 Hz, 2H), 5.80 (t, *J* = 5.3 Hz, 1H), 3.82 (dd, *J* = 11.9, 5.3 Hz, 1H), 3.56 (dd, *J* = 11.9, 5.5 Hz, 1H), 1.42 (s, 3H). ^13^C NMR (101 MHz, DMSO) δ 172.82, 153.09, 146.28, 140.32, 136.42, 136.22, 131.81, 131.50, 131.18, 130.85, 130.11, 128.80, 124.64, 123.96, 123.92, 123.50, 120.78, 115.05, 107.36, 69.14, 63.70, 30.36, 17.13.HRMS(ESI)*m/z* calcd for C_19_H_13_F_3_N_4_O_5_ [M+Na]^+^: 457.0736, Found: 457.0722.

##### 4-(4-(hydroxymethyl)-4-methyl-2,5-dioxo-3-(4-(trifluoromethyl)phenyl)imidazolidin-1-yl)-2-(trifluoromethyl)benzonitrile (**7i**)

White solid (190 mg, 66% yield). m.p. 161–163 °C; ^1^H NMR (400 MHz, DMSO) δ 8.43 (d, *J* = 8.4 Hz, 1H), 8.25 (d, *J* = 1.3 Hz, 1H), 8.12 (dd, *J* = 8.4, 1.5 Hz, 1H), 8.07 (d, *J* = 8.6 Hz, 2H), 7.75 (d, *J* = 8.6 Hz, 2H), 5.83 (t, *J* = 5.3 Hz, 1H), 3.85 (dd, *J* = 11.9, 5.2 Hz, 1H), 3.56 (dd, *J* = 11.9, 5.5 Hz, 2H), 1.44 (s, 3H). ^13^C NMR (101 MHz, DMSO) δ 172.96, 153.18, 137.72, 136.38, 131.48, 131.15, 129.97, 129.51, 128.70, 128.38, 126.49, 126.46, 125.28, 123.83, 123.79, 123.51, 122.58, 120.79, 115.07, 107.21, 68.82, 63.53, 17.09. HRMS(ESI)*m/z* calcd for C_20_H_13_F_6_N_3_O_3_ [M+H]^+^: 458.0939, Found: 458.0929.

##### methyl 4-(3-(4-cyano-3-(trifluoromethyl)phenyl)-5-(hydroxymethyl)-5-methyl-2,4-dioxoimidazolidin-1-yl)benzoate (**7j**)

White solid (70 mg, 50% yield). m.p. 139–141 °C; ^1^H NMR (400 MHz, DMSO) δ 8.37 (d, *J* = 8.4 Hz, 1H), 8.20 (d, *J* = 1.5 Hz, 1H), 8.08 (dd, *J* = 13.0, 5.1 Hz, 3H), 7.63 (d, *J* = 8.6 Hz, 2H), 5.76 (t, *J* = 5.3 Hz, 1H), 3.89 (s, 3H), 3.79 (dd, *J* = 11.8, 5.2 Hz, 1H), 3.47 (dd, *J* = 11.8, 5.4 Hz, 1H), 1.37 (s, 3H). ^13^C NMR (101 MHz, DMSO) δ 172.99, 165.58, 153.09, 138.35, 136.38, 131.14, 130.22, 129.98, 129.11, 128.67, 123.86, 123.81, 123.51, 120.79, 115.08, 107.18, 68.83, 63.59, 52.29, 17.16. HRMS(ESI)*m/z* calcd for C_21_H_16_F_3_N_3_O_5_ [M+H]^+^: 448.1120, Found: 448.1154.

##### *N*-(4-(3-(4-cyano-3-(trifluoromethyl)phenyl)-5-(hydroxymethyl)-5-methyl-2,4-dioxoimidazolidin-1-yl)phenyl)acetamide (**10**)

White solid (40 mg, 41% yield). m.p. 124–126 °C; ^1^H NMR (400 MHz, DMSO) δ 10.12 (s, 1H), 8.35 (d, *J* = 8.4 Hz, 1H), 8.18 (s, 1H), 8.06 (d, *J* = 8.5 Hz, 1H), 7.69 (d, *J* = 8.6 Hz, 2H), 7.36 (d, *J* = 8.6 Hz, 2H), 5.66 (t, *J* = 5.1 Hz, 1H), 3.73 (dd, *J* = 11.6, 4.9 Hz, 1H), 3.39−3.34 (m, 1H), 2.07 (s, 3H), 1.30 (s, 3H). ^13^C NMR (101 MHz, DMSO) δ 173.23, 168.49, 153.26, 139.51, 136.63, 136.32, 131.41, 131.09, 129.78, 127.77, 123.71, 123.65, 123.61, 123.54, 120.82, 119.49, 115.12, 106.90, 68.32, 63.27, 23.96, 17.05. HRMS(ESI)*m/z* calcd for C_21_H_17_F_3_N_4_O_4_ [M+Na]^+^: 469.1100, Found: 469.1100.

#### 4.2.5. 4-(3-(4-aminophenyl)-4-(hydroxymethyl)-4-methyl-5-oxo-2-thioxoimidazolidin-1-yl)-2-(trifluoromethyl)benzonitrile (**8**)

In a 10 mL round-bottomed flask, we added 4-(4-(hydroxymethyl)-4-methyl-3-(4-nitrophenyl)-5-oxo-2-thioxoimidazolidin-1-yl)-2-(trifluoromethyl)benzonitrile (**6h**) (200 mg, 0.444 mmol, 1.0 eq), iron (74.4 mg, 1.332 mmol, 3.0 eq) and HCl (0.222 mL, 0.888 mmol, 2.0 eq) in 5 mL ethanol to give a yellow suspension. The reaction was heated to 90 °C with stirring on for 4 h. The reaction mixture was filtered through celite. The mixture was concentrated by rotovap to give 4-(3-(4-aminophenyl)-4-(hydroxymethyl)-4-methyl-5-oxo-2-thioxoimidazolidin-1-yl)-2-(trifluoromethyl)benzonitrile (**8**) (170 mg, 91% yield) as a yellow solid. The crude product was used in the next step without further purification.

#### 4.2.6. N-(4-(3-(4-cyano-3-(trifluoromethyl)phenyl)-5-(hydroxymethyl)-5-methyl-4-oxo-2-thioxoimidazolidin-1-yl)phenyl)acetamide (**9**)

In a 25 mL round-bottomed flask, we added 4-(3-(4-aminophenyl)-4-(hydroxymethyl)-4-methyl-5-oxo-2-thioxoimidazolidin-1-yl)-2-(trifluoromethyl)benzonitrile (**8**) (170 mg, 0.404 mmol, 1.0 eq) in 5 mL THF to give a yellow solution. Acetyl chloride (0.035 mL, 0.485 mmol, 1.2 eq) was added. The reaction mixture was held at 0 °C with stirring on for 1 h. The mixture was concentrated by rotovap. The crude product was purified by column chromatography to give *N*-(4-(3-(4-cyano-3-(trifluoromethyl)phenyl)-5-(hydroxymethyl)-5-methyl-4-oxo-2-thioxoimidazolidin-1-yl)phenyl)acetamide (100 mg, 53.5% yield) as a yellow solid. m.p. 126–128 °C; ^1^H NMR (400 MHz, DMSO) δ 10.16 (s, 1H), 8.38 (d, *J* = 8.2 Hz, 1H), 8.14 (d, *J* = 1.2 Hz, 1H), 7.98 (dd, *J* = 8.3, 1.4 Hz, 1H), 7.72 (d, *J* = 8.8 Hz, 2H), 7.35 (d, *J* = 8.8 Hz, 2H), 5.82 (t, *J* = 5.1 Hz, 1H), 3.76 (dd, *J* = 11.6, 4.7 Hz, 1H), 3.39 (dd, *J* = 11.7, 5.6 Hz, 1H), 2.08 (s, 3H), 1.34 (s, 3H). ^13^C NMR (101 MHz, DMSO) δ 181.03, 174.00, 168.56, 139.92, 137.98, 136.24, 133.66, 131.26, 130.93, 129.94, 129.56, 127.47, 127.42, 123.49, 119.70, 119.53, 114.97, 108.46, 71.58, 63.56, 23.98, 16.76. HRMS(ESI)*m/z* calcd for C_21_H_17_F_3_N_4_O_3_S [M+Na]^+^: 485.0871, Found: 485.0886.

#### 4.2.7. General Procedure for the Synthesis of **11a-b**

In a 10 mL round-bottomed flask, we added compound **6f** or **7f** (1.0 eq) in 5 mL DCM to give a colorless solution. BBr_3_ (5.0 eq) was added. The reaction mixture was held at rt with stirring on for 2 h. Then, 5 mL NaHCO_3_(aq) was added. The aq layer was extracted with DCM. We combined the organic layers and washed them with brine. The organic was dried with Na_2_SO_4_, filt and conc to obtain a crude product. The crude product was purified by column chromatography to give **11a-b.**

##### 4-(4-(hydroxymethyl)-3-(4-hydroxyphenyl)-4-methyl-5-oxo-2-thioxoimidazolidin-1-yl)-2-(trifluoromethyl)benzonitrile (**11a**)

White solid (40 mg, 41% yield). m.p. 203–205 °C; ^1^H NMR (400 MHz, DMSO) δ 9.84 (s, 1H), 8.38 (d, *J* = 8.2 Hz, 1H), 8.12 (s, 1H), 7.97 (d, *J* = 6.5 Hz, 1H), 7.23 (d, *J* = 8.7 Hz, 2H), 6.89 (d, *J* = 8.7 Hz, 2H), 5.78 (t, *J* = 5.1 Hz, 1H), 3.79−3.68 (m, 1H), 3.44−3.34 (m, 1H), 1.32 (s, 3H). ^13^C NMR (101 MHz, DMSO) δ 181.09, 174.09, 157.94, 138.06, 136.20, 133.65, 131.23, 130.91, 130.62, 127.46, 127.41, 125.96, 123.50, 120.78, 115.88, 114.97, 108.41, 71.45, 63.52, 16.73. HRMS(ESI)*m/z* calcd for C_19_H_14_F_3_N_3_O_3_S [M+H]^+^: 422.0786, Found: 422.0783.

##### 4-(4-(hydroxymethyl)-3-(4-hydroxyphenyl)-4-methyl-2,5-dioxoimidazolidin-1-yl)-2-(trifluoromethyl)benzonitrile (**11b**)

White solid (60 mg, 78% yield). m.p. 110–112 °C; ^1^H NMR (400 MHz, DMSO) δ 9.77 (s, 1H), 8.34 (d, *J* = 8.4 Hz, 1H), 8.17 (s, 1H), 8.06 (d, *J* = 8.5 Hz, 1H), 7.23 (d, *J* = 8.8 Hz, 2H), 6.86 (d, *J* = 8.8 Hz, 2H), 5.63 (t, *J* = 5.1 Hz, 1H), 3.71 (dd, *J* = 11.5, 4.9 Hz, 1H), 3.34 (d, *J* = 5.4 Hz, 1H), 1.27 (s, 3H). ^13^C NMR (101 MHz, DMSO) δ 173.36, 157.63, 153.33, 136.65, 136.29, 131.40, 131.08, 130.80, 129.75, 123.91, 123.65, 123.61, 123.52, 120.80, 115.80, 115.14, 106.81, 68.13, 63.12, 16.95. HRMS(ESI)*m/z* calcd for C_19_H_14_F_3_N_3_O_4_ [M+H]^+^: 406.1015, Found: 406.1019.

#### 4.2.8. 5-nitro-3-(trifluoromethyl)pyridin-2-ol (**13**)

In a 500 mL round-bottomed flask, we added added 3-(trifluoromethyl)pyridin-2-ol (**12**) (9 g, 55 mmol, 1.0 eq) in 40 mL sulfuric acid to give an orange solution. The reaction mixture was cooled to 0 °C with stirring on. HNO_3_ (19 mL, 414 mmol, 7.5 eq) in 20 mL sulfuric acid was added slowly at 0 °C for 1 h. The reaction mixture was held at room temperature with stirring on for 6 hr. The mixture was poured into 500 mL ice-water with stirring on for 2 h. The reaction mixture was filtered through a Buchner funnel and washed with water to give 5-nitro-3-(trifluoromethyl)pyridin-2-ol (**13**) (8.6 g, 75% yield) as a white solid. m.p. 160–162 °C; ^1^H NMR (400 MHz, DMSO) δ 13.48 (s, 1H), 8.95 (d, *J* = 3.0 Hz, 1H), 8.47 (d, *J* = 2.6 Hz, 1H). ^13^C NMR (101 MHz, DMSO) δ 157.98, 142.35, 134.16, 134.11, 128.69, 126.01, 123.32, 120.61, 117.91, 117.31, 116.99, 116.68, 116.37.

#### 4.2.9. 2-bromo-5-nitro-3-(trifluoromethyl)pyridine (**14**)

In a 100 mL round-bottomed flask, we added 5-nitro-3-(trifluoromethyl)pyridin-2-ol (**13**) (6 g, 28.8 mmol, 1.0 eq), phosphorus oxybromide (24.80 g, 86 mmol, 3.0 eq) and DMF (0.335 mL, 4.32 mmol, 0.15 eq) to give a yellow suspension. The reaction was heated to 110 °C with stirring on for 3 h. The reaction mixture was added into 200 g ice portion-wise, then, we adjusted the pH to 7. The aq layer was extracted with EA. We combined the organic layers and washed them with water and brine. The organic was dried with Na_2_SO_4_, filt and conc to give a crude product. The crude product was purified by column chromatography to give 2-bromo-5-nitro-3-(trifluoromethyl)pyridine (**14**) (7.61 g, 97% yield) as a light-yellow solid. m.p. 38–40 °C; ^1^H NMR (400 MHz, CDCl_3_) δ 9.36 (d, *J* = 1.2 Hz, 1H), 8.76 (d, *J* = 1.6 Hz, 1H). ^13^C NMR (101 MHz, CDCl_3_) δ 147.29, 145.58, 142.82, 131.60, 131.54, 131.49, 131.44, 129.28, 128.93, 128.58, 128.24, 125.24, 122.52, 119.80, 117.08.

#### 4.2.10. 6-bromo-5-(trifluoromethyl)pyridin-3-amine (**15**)

In a 250 mL round-bottomed flask, we added 2-bromo-5-nitro-3-(trifluoromethyl)pyridine (**14**) (7.5 g, 27.7 mmol, 1.0 eq), iron (5.41 g, 97 mmol, 3.5 eq), and HOAc (23.77 mL, 415 mmol, 15 eq) in 25 mL ethyl acetate to give a black suspension. The reaction was heated to 65 °C with stirring on for 2 h. sat. Na_2_CO_3_ (aq) was added to adjust the pH to 10. The reaction mixture was filtered through a Buchner funnel. The aq layer was extracted with EA. We combined the organic layers and washed them with water and brine. The organic was dried with Na_2_SO_4_, filt and conc to give 6-bromo-5-(trifluoromethyl)pyridin-3-amine (**15**) (6.1 g, 91% yield) as a yellow solid. The product was used in the next step without further purification. ^1^H NMR (400 MHz, CDCl_3_) δ 7.97 (s, 1H), 7.26 (s, 1H), 3.81 (s, 2H). ^13^C NMR (101 MHz, CDCl_3_) δ 141.94, 138.79, 127.51, 127.19, 125.38, 123.56, 121.82, 121.77, 121.72, 120.85.

#### 4.2.11. 5-amino-3-(trifluoromethyl)picolinonitrile (**16**)

In a 250 mL round-bottomed flask, we added 6-bromo-5-(trifluoromethyl)pyridin-3-amine (**15**) (6 g, 24.90 mmol, 1.0 eq) and copper(I) cyanide (2.56 g, 28.6 mmol, 1.2 eq) in 60 mL NMP to give a black suspension. The reaction vessel was purged with nitrogen. The reaction mixture was heated to 160 °C with stirring on for 2 h. Then, 200 mL 25% EDA (aq) was added. The aq layer was extracted with EA. We combined the organic layers and washed them with 25% EDA (aq) and brine. The organic was dried with Na_2_SO_4_, filt and conc to give a crude product (7g). The crude product was purified by column chromatography to give 5-amino-3-(trifluoromethyl)picolinonitrile (**16**) (4.2 g, 90% yield) as a yellow solid. M.p. 148–150 °C; ^1^H NMR (400 MHz, DMSO) δ 8.17 (d, *J* = 1.7 Hz, 1H), 7.27 (d, *J* = 1.9 Hz, 1H), 6.97 (s, 2H). ^13^C NMR (101 MHz, DMSO) δ 147.96, 139.37, 130.34, 130.02, 129.70, 129.38, 126.28, 123.56, 120.84, 118.12, 116.01, 115.07, 115.02, 114.97, 114.93, 112.92.

#### 4.2.12. 5-isothiocyanato-3-(trifluoromethyl)picolinonitrile (**17**)

In a 100 mL round-bottomed flask, we added 5-amino-3-(trifluoromethyl)picolinonitrile (**16**) (1.0 g, 5.34 mmol, 1.0 eq) in 20 mL DCM and 10 mL water to give an orange solution. Thiophosgene (0.451 mL, 5.88 mmol, 1.1 eq) was added. The reaction mixture was held at rt with stirring on for 16 h. The aq layer was extracted with DCM. We combined the organic layers and washed them with water and brine. The organic was dried with Na_2_SO_4_, filt and conc to give a crude product. The crude product was purified by column chromatography to give 5-isothiocyanato-3-(trifluoromethyl)picolinonitrile (**17**) (930 mg, 76% yield) as a yellow oil. The product was used in the next step without further purification. m.p. 30–32 °C; ^1^H NMR (400 MHz, CDCl_3_) δ 7.85 (d, J = 2.4 Hz, 1H), 8.72 (d, J = 2.4 Hz, 1H); ^13^C NMR (100 MHz, CDCl_3_) δ 113.61, 121.04, 127.41, 130.38, 131.44, 133.55, 144.75, 150.30.

### 4.3. Bioassays

#### 4.3.1. Cell Preparation

C2C12 cells were provided and certified by the Cell Bank at Shanghai Institutes for Biological Sciences, Chinese Academy of Sciences and confirmed as being negative for mycoplasma contamination. C2C12 were cultured in phenol red-free HG–DMEM medium (Life Technologies, Carlsbad, CA, USA) supplemented with 10% (*v*/*v*) fetal bovine serum, 50 IU/mL penicillin, 50 μg /mL streptomycin and 1% sodium pyruvate. Cells were maintained at 37 °C in a 5% CO_2_ incubator and seeded onto 10 cm cell culture dishes before transfection. After overnight culture, the cells were transiently co-transfected with AR-expressing plasmid (pSVAR0) and luciferase reporter gene vector (MMTV-Luc) using FuGENE^®^ HD Transfection Reagent (Promega, Madison, WI, USA), while cells reached 80–90% confluence. After 18 h, the transfected cells were distributed to 384-well plates at a density of 15,000 cells per well and incubated for a further 6 h at 37 °C before compound treatment.

#### 4.3.2. Antagonist Assay

Enzalutamide was used as a positive control. The antagonist activity of testing compounds was measured by Steadylite plus the Reporter Gene Assay System (PerkinElmer, Boston, MA, USA) according to the manufacturer’s instructions. In brief, after incubation for 6 h as mentioned above, 5 μL testing of compounds diluted in culture medium with eight different working concentrations (384 pM to 30 μM; enzalutamide (128 pM–10 μM)) was added to the cell well followed by 5 μL DHT (final concentration as EC_80_). After 24 h incubation, Steadylite reagent (50 μL, equal volume) was introduced, gently shaken for 2 min and kept at room temperature for 15 min before luminescence 384 measurement on an EnSpire multilabel plate reader (PerkinElmer).

#### 4.3.3. Cytotoxicity

CellTiter-Glo^®^ 2.0 Assay (Promega) was applied to assess cytotoxicity. In brief, cells were seeded onto 384-well plates at a density of 1500 cells per well and incubated for 24 h. Ten microliters of testing compounds diluted in culture medium was added and reacted for 24 h. CellTiter-Glo reagent was then introduced and the luminescence 384 was measured as above.

#### 4.3.4. Anti-Proliferative Activity Assay

LNCaP cells were cultured in RPMI-1640 medium (Life Technologies, Carlsbad, CA, USA) supplemented with 10% (*v*/*v*) fetal bovine serum, 1% sodium pyruvate and 1% L-Glutamine. The cells were seeded into 96-well plates at a density of 8 × 104 cells per well. After 24 h incubation, the culture medium was removed and the cells were cultivated in the medium supplemented with 2% (*v*/*v*) fetal bovine serum for 2 days and then treated with different concentrations of test compounds for 3 days. Cell viability was measured with the CCK-8 kit (Dojindo).

### 4.4. Molecular Docking

All docking procedures were completed with programs as implemented in Schrodinger Suite.

The crystal structure of the androgen receptor (PDB: 2OZ7) was prepared in Protein Preparation Wizard with all water molecules deleted and bond orders assigned. The 2D structures of 19b, enzalutamide and apalutamide subjected to LigPrep and possible tautomeric states at pH 7.0 ± 2.0 were generated using Epik. Induced-fit docking was performed using the program Induced Fit and briefly consisted of several steps: first, the initial docking was performed using Glide, and then the sampling and minimization of sidechain residues within 5 Å of the docked ligand was achieved using Prime, followed by re-docking using Glide.

In the initial docking, the receptor van der Waals radii scaling was set at 0.50 and the ligand van der Waals radii was scaled at 0.5 to soften the potentials. Finally, the binding energy for each output pose was estimated as the IFDScore.

## Data Availability

Not applicable.

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
