# Peer review of "Synthesis of Hydantoin Androgen Receptor Antagonists and Study on Their Antagonistic Activity"

_molecules, 2022, doi:10.3390/molecules27185867_

Round 1

Reviewer 1 Report

1) Author could provide % yield for reactions in synthesis scheme

2) Typo error - Line 59 and 60. (CF3 instead SF5)

3) Table 1 and Table 2 SD values are missing.

4) Author could provide explanation about % Efficacy and how it's calculated.

5) Please explain, For Enzalutamide different IC50 values are shown in table 1 and table 3. 

Author Response

Dear review

   Thank you very much for your comments and suggestions on our work. We have revised our manuscript according to the comments and suggestions.

Reviewer 2 Report

In the current work “Synthesis of Hydantoin androgen Receptor Antagonists and Study on Their Antagonistic Activity” by Longjun Ma et al, authors reported new series of AR Antagonists based on the Hydantoin scaffold. Before considering this work for publishing in Molecules, some issues need to be addressed.

- In the Anti-proliferative activity assay, it is not clear what did the authors measure!!

Are the value presented in table 4 for the proliferation, growth inhibition % or IC50??

What are the concentrations used in this assay?

Please revise the results and present it correctly.

- I wonder if the authors cold assess the cytotoxicity of this molecules towards normal cells!

- There are many issues in the docking study:

It lacks the validation of the used protocol.

Why did authors selected this pdb that has structurally non-related co-crystalzed ligands.

How the co-crystalzed ligand interacts within the active site, and what are the critical residuals that authors mentioned in the discussion.

The docking Figure should be no. 3 not no.1.

The quality of the docking Figure is low.

- The language of the manuscripts needs revision and editing.

- Also, there a lot ot typos and mistakes.

§  In abstract: “Hydroxymethylthiohydantoin, hydroxymethylthiohydantoin, and hydantoin containing 14 a pyridine group”

§  In the introduction: Please correct “As first generation nonsteroidal AR antagonists, bicalutamide diminish androgenic effects” to “As a first generation nonsteroidal AR antagonist, bicalutamide diminishs androgenic effects”

§  In the introduction: Please correct “Two agents are potent AR antagonist” to “The two agents are potent AR antagonists”

§  I recommend change “The transformation strategy” expression.

§  Please correct “around the thiohydanthoin and hydantoin scaffold” to “around the thiohydanthoin and hydantoin scaffolds”.

§  Please correct “Various compound (±)-6a-k, (±)-7a-j were synthesized” to “Various compounds (±)-6a-k, (±)-7a-j were synthesized”.

§  Please correct “Then, thiohydantoins (±)-9 were transformed” to “Then, thiohydantoin (±)-9 was transformed”.

- I suggest reducing the size of Scheme 2.

Author Response

(The authors gave the same response as above.)

Reviewer 3 Report

The manuscript entitled " Synthesis of Hydantoin androgen Receptor Antagonists and Study on Their Antagonistic Activity" is an interesting attempt to gain more information about novel enzalutamide analogs and their antioxidant activity.

 The authors synthesized a series of new derivatives, then they examined the AR antagonist activity of the representative compounds with a luciferase gene reporter assay in mouse myoblast CV-1 cells. Anti-proliferative activity of four compounds was also predicted. The structures of new compounds were confirmed by HRMS, 1H NMR and 13C NMR spectra.

The overall composition of the manuscript is good, with a very nice introduction to the topic. The quality of the figures and tables is also good. The paper is written in a clear and understanding manner and the results are properly presented. The synthesized compounds are sufficiently characterized from the chemical point of view.

Overall I have find this manuscript suitable for publication in “Molecules”, subject to the comments below:

1. I would suggest to include a full interpretation of the 13C NMR spectra, assigning the signals to the appropriate carbon atoms in the compounds.

2. I also did not find any supplementary materials that should contain spectra made.

Author Response

(The authors gave the same response as above.)

Round 2

Reviewer 2 Report

The manuscript has been improved.